# Cryo-EM study of start codon selection during archaeal translation initiation

Pierre-Damien Coureux[1], Christine Lazennec-Schurdevin[1], Auriane Monestier[1], Eric Larquet[2], Lionel Cladière[1,†], Bruno P. Klaholz[3], Emmanuelle Schmitt[1,*] & Yves Mechulam[1,*]

Eukaryotic and archaeal translation initiation complexes have a common structural core comprising e/aIF1, e/aIF1A, the ternary complex (TC, e/aIF2-GTP-Met-tRNA$_i^{Met}$) and mRNA bound to the small ribosomal subunit. e/aIF2 plays a crucial role in this process but how this factor controls start codon selection remains unclear. Here, we present cryo-EM structures of the full archaeal 30S initiation complex showing two conformational states of the TC. In the first state, the TC is bound to the ribosome in a relaxed conformation with the tRNA oriented out of the P site. In the second state, the tRNA is accommodated within the peptidyl (P) site and the TC becomes constrained. This constraint is compensated by codon/anticodon base pairing, whereas in the absence of a start codon, aIF2 contributes to swing out the tRNA. This spring force concept highlights a mechanism of codon/anticodon probing by the initiator tRNA directly assisted by aIF2.

[1] Laboratoire de Biochimie, Ecole polytechnique, CNRS, Université Paris-Saclay, 91128 Palaiseau cedex, France. [2] Laboratoire de Physique de la Matière Condensée, Ecole polytechnique, CNRS, Université Paris-Saclay, 91128 Palaiseau cedex, France. [3] Department of Structural Biology and Genomics, Institute of Genetics and Molecular and Cellular Biology, Centre National de la Recherche Scientifique (CNRS)/Institut National de la Santé et de la Recherche Médicale (INSERM)/Université Louis Pasteur, BP 10142, 67404 Illkirch, France. * These authors contributed equally to this work. † Present address: Station Biologique de Roscoff, Place Georges Teissier, 29680 Roscoff, France. Correspondence and requests for materials should be addressed to P.-D.C. (email: pierre-damien.coureux@polytechnique.edu) or to E.S. (email: emmanuelle.schmitt@polytechnique.edu).

In all living cells, translation initiation allows accurate selection of the initiation codon on a messenger RNA, which then defines the reading frame of the protein to be synthesized. In eukaryotes, this process is highly complicated and is the target of many regulations[1–3]. The mechanism begins with the assembly of the ternary complex (TC), consisting of initiator methionyl-transfer RNA (Met-tRNA$_i$) anchored to the GTP-bound form of the heterotrimeric initiation factor 2 (eIF2). The TC then binds to the small ribosomal subunit (40S) in the presence of two small initiation factors eIF1 and eIF1A. TC binding is assisted by eIF5 (ref. 4) and by the multimeric factor eIF3 (ref. 5). eIF5 favours reversible GTP hydrolysis on eIF2. In the presence of factors belonging to the eIF4 family, the pre-initiation ribosomal complex (PIC) is recruited near the 5′-capped end of the messenger RNA (mRNA) and scans the mRNA to search for an AUG codon in an appropriate context. AUG recognition stops scanning, leads to factor release and to the assembly of an elongation-proficient 80S complex through large subunit joining, with the help of eIF5B. A large amount of data from genetics and biochemistry has been collected to decipher the molecular steps ensuring the correct selection of the AUG codon on a messenger RNA[6–10]. In particular, isolation of yeast mutants able to initiate translation on a non-AUG codon highlighted the key roles of eIF1 and eIF2 in the selection of the start codon. In the current model, AUG recognition triggers a conformational change of the 40S subunit, which leads to eIF1 departure, and then to the release of inorganic phosphate (Pi) from eIF2 (ref. 7). In its GDP-bound form, eIF2 loses its affinity for the initiator tRNA and therefore leaves the 40S subunit[11,12]. Recent structural descriptions of eukaryotic translation initiation complexes (ICs) have highlighted some of the structural changes that lead to a closed state with the initiator tRNA base-paired to the start codon[13–16]. These structures have shown how accommodation of the initiator tRNA at the P site, from a $P_{OUT}$ to a $P_{IN}$ position[7,17], is stabilized by eIF1A and is accompanied by a closure of the mRNA channel in the 40S subunit as well as by the destabilization of eIF1 binding. However, these structures do not give any clue on how the departure of eIF1 triggers the release of the Pi group from eIF2 and thereby the release of eIF2-GDP. Moreover, the absence of contact between the central GTP-binding γ-subunit of eIF2 and the ribosome remains intriguing according to hydroxyl radical probing experiments that suggest interaction of domain III of eIF2γ with h44 in rRNA[18].

In archaea, long-range scanning does not occur because of the occurrence of Shine-Dalgarno (SD) sequences or of very short 5′ untranslated region on mRNA. However, archaeal translation initiation uses three initiation factors, aIF1, aIF1A and aIF2, homologous to their eukaryotic counterparts. This suggests that the archaeal factors fulfill similar functions as the eukaryotic factors[19]. Such a parallel relies on the idea that the detection of the start codon in archaea proceeds through a local scanning mechanism within a short mRNA region compatible with maintenance of SD:antiSD pairing. This leads to the definition of a structural core, common to eukaryotes and archaea, that controls translation initiation accuracy. This core is made of the small ribosomal subunit, the mRNA, the Met-tRNA$_i^{Met}$ and the three factors e/aIF1, e/aIF1A and e/aIF2.

To further address the role of e/aIF2, we have analysed the full archaeal 30S initiation complex (30S IC) by cryo-EM. We isolated two conformational states of the 30S IC at 5.34 and 7.5 Å overall resolutions, respectively. All initiation factors are visible, even though at a generally worse resolution. In the first complex, the tRNA is oriented out of the peptidyl (P) site in a standby position (the complex is therefore called IC0–$P_{REMOTE}$) and the hetero-trimeric aIF2 strongly interacts with the 30S subunit and with aIF1. In the second complex, the initiator tRNA is accommodated

within the P site (the complex is therefore called IC1–$P_{IN}$) and partially releases interactions with aIF2, reflecting the structural transition from the standby position to the decoding position. Comparison of IC0–$P_{REMOTE}$ and IC1–$P_{IN}$ shows that the TC is constrained when the tRNA is engaged in the P site. However, on start codon recognition this constraint is compensated by codon/anticodon base pairing. In the absence of a start codon, aIF2 contributes to swing out the tRNA from the P site. This spring force concept provides a novel view of the role of e/aIF2 in start codon selection. Moreover, we observe contacts between aIF1 and the aIF2γ switch regions known to control its nucleotidic state. This gives a structural framework to explain how aIF1 departure provokes the release of Pi from aIF2. According to previous biochemical and genetic studies in eukaryotic systems[7,9,18], the features of the two novel states of the archaeal core IC are likely to be relevant for eukaryotic translation initiation.

## Results

**Assembly and structure of the full archaeal IC.** We assembled the full IC using archaeal 30S subunits from *Pyrococcus abyssi* (Pa-30S), initiation factors Pa-aIF1, Pa-aIF1A, the ternary complex (Met-tRNA$_i^{Met}$ A$_1$-U$_{72}$ variant from *Escherichia coli* (Supplementary Fig. 1a), GDPNP and Pa-aIF2 composed of three subunits α, β and γ) and a synthetic 26 nucleotide long mRNA corresponding to the natural start region of the mRNA encoding the elongation factor aEF1A from *P. abyssi*, which contains a strong Shine-Dalgarno sequence (A$_{(-17)}$UUU**GGAGGUGAU** UUAAA$_{(+1)}$**UG**CCAAAG$_{(+9)}$). The complex was purified by affinity chromatography using N-terminally tagged versions of Pa-aIF1, Pa-aIF2α and Pa-aIF2β (Supplementary Fig. 1b). An excess of mRNA, aIF1A and TC was added before spotting onto the grids for Cryo-EM analysis. Cryo-EM images were collected on an FEI Cs-corrected Titan Krios microscope operated at 300 kV (Supplementary Fig. 1c). After image processing and three-dimensional (3D) classification using RELION[20], the structures of two complexes were identified (Figs 1 and 2 and Supplementary Fig. 2). In one complex, the tRNA is totally out of the P site and is therefore referred to as IC0–$P_{REMOTE}$. The other structure, featuring a tRNA accommodated in the P site is named IC1–$P_{IN}$. The two electron density maps were obtained at overall resolutions of 5.34 and 7.5 Å for IC0–$P_{REMOTE}$ and IC1–$P_{IN}$, respectively, according to the 0.143 FSC criterion[21] (Supplementary Fig. 2b, Fig. 2 and Table 1).

As already observed for other ICs[22], local resolution is best in the core of the 30S subunit and decreases as distance from the 30S increases (Fig. 2 and Table 1).

Model construction was mainly based on the placement of known structures of archaeal homologous components (Supplementary Fig. 3 and 'Methods' section). In both eukaryotes and archaea, e/aIF2 is an αβγ heterotrimer. Structural data obtained with the archaeal versions of the protein have shown that the heterotrimer contains a rigid unit made of the γ-subunit, domain 3 of the α-subunit (aIF2α-D3) and the N-terminal helix of the β-subunit. Onto this rigid unit are anchored two mobile wings corresponding to domains 1 and 2 of α (aIF2α-D12) and to the main domain of aIF2β (refs 23–26). These structural features were taken into account for aIF2 fitting. In both IC0–$P_{REMOTE}$ and IC1–$P_{IN}$, aIF1A and aIF1 are present in the A site and in front of the P site, respectively, similarly to their eIF1A and eIF1 orthologues in the previous structures of eukaryotic complexes[15,16,22,27] (Fig. 3). The mRNA is clearly visible in the electron density. The interaction of the SD sequence with the anti-SD was modelled and the start codon was positioned within the P site (Supplementary Fig. 4a). The mRNA is not visible from

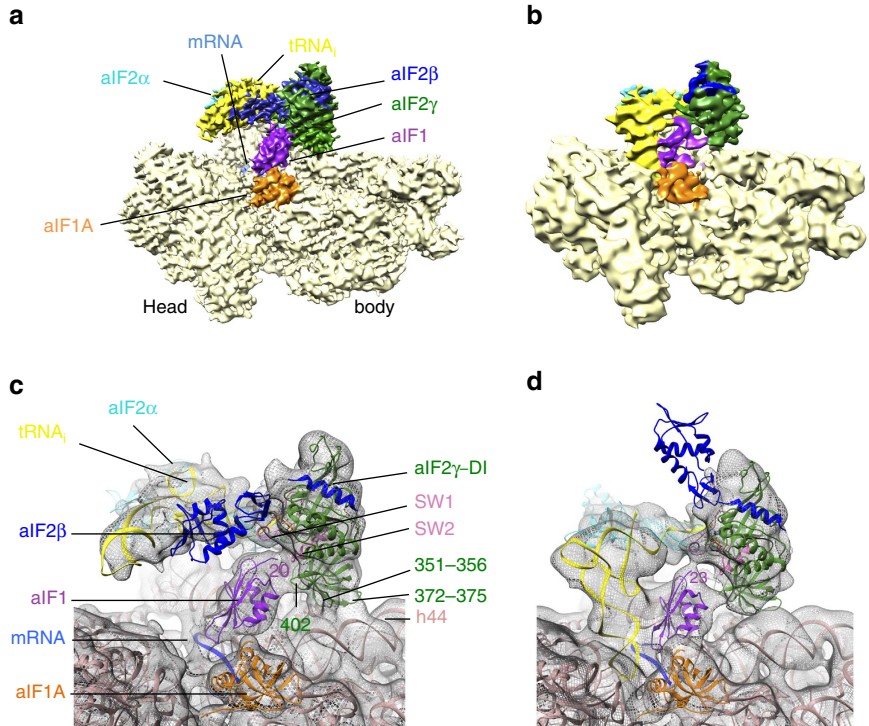

**Figure 1 | Cryo-EM maps of archaeal initiation complexes**. (**a**) Cryo-EM map of the IC0–P$_{REMOTE}$ complex at 5.34 Å resolution. Regions of the map are coloured by components to show the 30S subunit (pale yellow), aIF1A (orange), aIF1 (magenta), Met-tRNA$_i^{Met}$ (yellow), mRNA (light blue), aIF2γ (green), aIF2β (blue) and aIF2α (cyan). The same colour code is used in all the figures. (**b**) Cryo-EM map of the IC1–P$_{IN}$ complex at 7.5 Å resolution. (**c**) P site region in IC0–P$_{REMOTE}$. The cryo-EM map has been low-pass filtered to 8.5 Å resolution. The models used to account for the electron density correspond to the structures of *Pyrococcus furiosus* 30S (PDB entry 4V6U; ref. 48), of *Sulfolobus solfataricus* TC (PDB entry 3V11; ref. 23), of Pa-aIF1A (PDB entry 4MNO) and of *Methanocaldococcus jannaschii* aIF1 (PDB entry 4MO0; Supplementary Fig. 3). Note that the residue numbering used in the text refers to that of the fitted components. Switch 1 and 2 regions in aIF2γ are coloured in light pink. GDPNP is shown in sticks. This view also highlights contacts between aIF1 and aIF2γ. One contact involves the β20–β21 loop (residues 402–404) of aIF2γ with the aIF1 α2 helix. A second area of density connects the two switch regions of aIF2γ and the N-terminal domain of aIF1. (**d**) P site region in IC1–P$_{IN}$. The density map has been low-passed filtered to 12 Å resolution. Note that positioning of the core domain of aIF2β is only tentative.

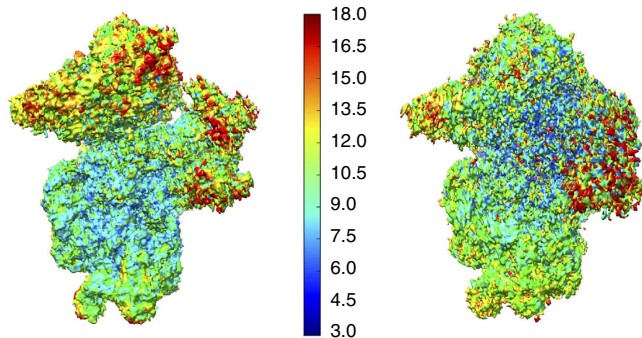

**Figure 2 | Local resolutions of ICO–P$_{REMOTE}$ and IC1–P$_{IN}$ maps.** Left: surface of IC0–P$_{REMOTE}$ map coloured according to local resolution (see Supplementary Methods) using the colour scale indicated. Right: same for IC1–P$_{IN}$.

base $+5$ to $+9$ at the 3′ extremity, as in the yeast 48S complex (py48S)[14]; bases $-17$ and $-16$ at the 5′ end are not visible.

**IC0–P$_{REMOTE}$ shows a new network of interactions around aIF2.** In IC0–P$_{REMOTE}$, all actors of translation initiation are bound to the ribosome but the initiator tRNA is located totally outside of the P site (Fig. 1a,c). The atomic model corresponding to the archaeal TC[23] could be readily fitted in the electron density. Therefore, on the 30S subunit, the initiator tRNA is bound to aIF2 in a manner

similar to that in the free TC, without constraints induced by the ribosome. As compared with the crystal structure, only slight rotations of the rigid unit made of domains 1 and 2 of aIF2α (aIF2α–D12) are observed, accompanied by a minor movement of the tRNA (Supplementary Fig. 3b).

Within the heterotrimeric aIF2, the γ-subunit is tightly bound to the 30S subunit, mainly held by two contact points (Figs 1 and 4). One interaction involves domain III of aIF2γ (aIF2γ–DIII). More precisely, the carboxy (C)-terminal part of the long loop between β16 and β17 as well as the loop between β18 and β19 overhang the top of the h44 bulge (Fig. 4b). Accordingly, a distortion of the helical structure of h44 is observed and the local structure of h44 had to be remodelled (Fig. 4c). Also, the C terminus of ribosomal protein uL41 points towards aIF2γ–DIII (Fig. 4b). A second contact between aIF2 and the 30S subunit involves the long β-hairpin (β10–β11) of aIF2γ–DII and the groove formed by the rRNA h27 loop on one side and by the extremity of h24 on the other side. Notably, this region of the rRNA links the decoding centre and the platform. This contact gives sense to the presence of the β10–β11 hairpin in all e/aIF2γ, whereas it is absent in elongation factors e/aEF1A, despite strong structural homology between the two proteins[28]. The position of aIF2α–D3 with respect to aIF2γ corresponds to that observed in the TC and in other aIF2 structures[26,29,30]. aIF2α–D3 is sitting on the platform close to the tip of h23 and h24 bulges. Moreover, the C terminus of aIF2α points towards h24 (Fig. 4b). Interestingly, the length of the e/aIF2α C-terminal tail varies from species to species. In eukaryotes, this tail was shown to negatively influence tRNA

**Table 1 | Local resolution of initiation factors.**

| Initiation factor | IC0-P$_{REMOTE}$ | IC1-P$_{IN}$ |
| --- | --- | --- |
| aIF1A | 6.0 | 9.3 |
| aIF1 | 6.9 | 8.5 |
| tRNA | 8.7 | 10 |
| aIF2α–D3–Nterβ | 8.3 | 15.6 |
| aIF2α–D12 | 15 | 15 |
| aIF2β | ND | ND |

ND, not determined. The table shows the local resolutions in Å of the components of the initiation complexes as determined with RELION using a soft mask (with a 15-pixel soft edge) surrounding the region of interest as described[59].

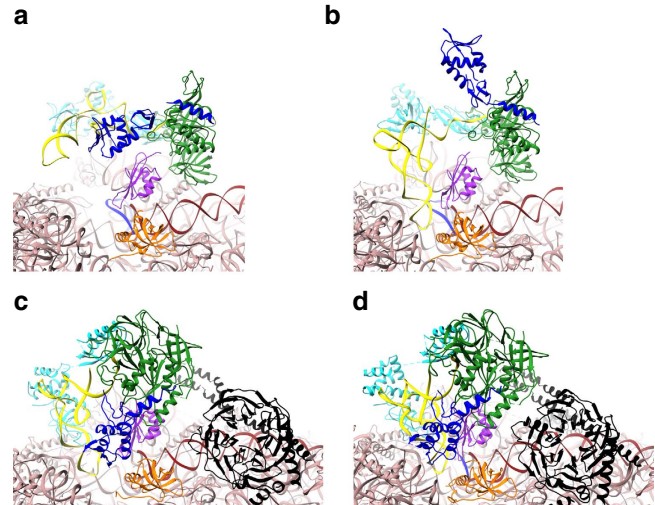

**Figure 3 | Comparison of archaeal and eukaryotic initiation complexes.** (**a**) IC0-P$_{REMOTE}$. (**b**) IC1-P$_{IN}$. (**c**) Eukaryotic open complex (PDB entry 3JAQ; ref. 14). (**d**) Eukaryotic closed complex (PDB entry 3JAP; ref. 14). Colour code for the initiation factors is the same as that used in Fig. 1 except that h44 is in brown and the small ribosomal subunit in pink. Colour code for eukaryotic factors corresponds to that used for archaeal orthologues. The eIF3 subunits are in black. The figure shows that archaeal aIF1 and aIF1A occupy similar positions as their eukaryotic orthologues eIF1 and eIF1A. In contrast, the positions of the archaeal TC in either IC0-P$_{REMOTE}$ or IC1-P$_{IN}$ differ from the positions of the eukaryotic TC observed in the eukaryotic structures.

binding, and it was suspected that this negative effect might be cancelled on interaction of the TC with the ribosome[23,31]. In this context, it is notable that a contact between the C-terminal tail of eIF2α and h24 can be inferred from the position of aIF2α-D3 observed here. The release of such a contact may help the release of eIF2 from the tRNA on start codon recognition.

On the side of aIF2α–D12, electron density suggests contacts with ribosomal protein eS1. Notably, in the case of py48S, eIF2α–D12 was located in the E site, close to the −2 and −3 positions of mRNA (Fig. 3c,d). This position is suitable for a participation of eIF2α in sensing the favourable environment of the correct start codon during mRNA scanning[32]. The different orientation of aIF2α–D12 observed here is possibly related to the pre-positioning of the 30S subunit on the mRNA due to the recognition of the SD sequence.

Finally, the electron density also suggests interaction of the αβ domain of aIF2β with the D-stem and D-loop of the tRNA (Figs 1c and 4d). Such a position of aIF2β with respect to the tRNA and to aIF2γ is similar to that observed in eukaryotic translation ICs[14] (Fig. 3).

Hence, on the whole, the initiator tRNA appears to be wrapped up by the three aIF2 subunits leaving the anticodon loop free of interaction (Figs 1 and 4d). Notably, the observed position of the ternary complex on the 30S subunit prevents uncontrolled tRNA access to the ribosomal E site.

In front of the aIF2γ–DIII domain, aIF1 occupies a position corresponding to that previously described for eIF1 in eukaryotic ICs, on h44 close to the P site[15,16,22] (Figs 1 and 3). As compared with the structure of the isolated factor, the conformation of the basic L1 loop had to be remodelled to account for the electron density. The loop comes close to the P site codon on the mRNA, suggesting that aIF1 contributes to its stabilization in the mRNA channel. As compared with eukaryotic complexes[13,14,22], a novel feature of the present structure is that aIF1 and aIF2 interact with each other (Figs 1b and 3). One contact between both initiation factors involves the β20–β21 loop of aIF2γ–DIII (402–404) with the top of the aIF1 α2 helix and the following loop 91–95. A supplementary density at the N terminus of aIF1 corresponds to a part of the 23 N-terminal residues missing in the aIF1 crystal structure, and shows the occurrence of a contact between the N-terminal domain of aIF1 and the two switch regions of aIF2γ. Finally, the electron density of aIF1 in IC0—P$_{REMOTE}$ is as strong as that observed for the ribosome, reflecting its tight binding (Figs 1 and 2).

aIF1A is positioned within the A site as its eIF1A orthologue in eukaryotic PIC[15,16,22] (Fig. 3 and Supplementary Fig. 4b). Interactions between h44 and the tight turn 34–36 are visible. To avoid clashes with aIF1A, bases 1447 and 1448 (1492 and 1493 in *E. coli* numbering) must flip out from h44, as previously observed for eukaryotic complexes[15,16,22]. Notably, the channel leading to the P site, corresponding to that occupied by the functionally important N-terminal extremity of eIF1A[33] in

the py48S complex[22], is empty in the present structure (Supplementary Fig. 4b). The N-terminal tail of Pa-aIF1A is 22-residues long, 10 residues shorter than that of the eukaryotic factor. Residues 13 to 22 form an elongated peptide with its extremity facing the h32–h34 junction in the 30S head but the first 12 residues are not visible within the electron density.

**IC1–P$_{IN}$ shows a constrained state of the TC.** In IC1–P$_{IN}$, the initiator tRNA is positioned in the P site (Fig. 1b,d). The density is consistent with the interaction of the anticodon with the start codon on mRNA whereas the 3′ methionylated extremity remains bound to aIF2γ. The structure of the P site initiator tRNA as observed in the *Thermus thermophilus* 70S complex[34] could not be readily rigid-body fitted in the density. Indeed a bending of the tRNA molecule at the level of the hinge region was necessary (Fig. 5b and Supplementary Fig. 5c). The acceptor end was also remodelled to fit the 3′ methionyl-adenosine in its pocket on aIF2γ–DII (Supplementary Fig. 6a). Notably, aIF2γ moves only very slightly in IC1–P$_{IN}$ as compared with IC0–P$_{REMOTE}$ and remains bound to h44 via its domain III (Figs 1 and 3a,b). Hence, in IC1–P$_{IN}$, the tRNA molecule interacts with aIF2 in a manner different from that in free TC or in IC0–P$_{REMOTE}$ (Fig. 1). Overall, both the tRNA structure and the tRNA:aIF2γ interface are constrained in IC1–P$_{IN}$ (Fig. 5a). Remarkably, induced structural constraints have already been observed during accommodation of the EF1A:tRNA complex in the A site on codon recognition[35,36] (Supplementary Fig. 6). The remodelling of the TC is accompanied by the loosening of some contacts between aIF2 and the 30S subunit such as those involving the long L2 loop of aIF2γ–DII and the aIF2α–D3 domain (Supplementary Fig. 5). Moreover, concomitant with its adjustment within the P site, the contacts of the tRNA with aIF2α–D3 are lost. Accordingly, the electron density of aIF2α–D3

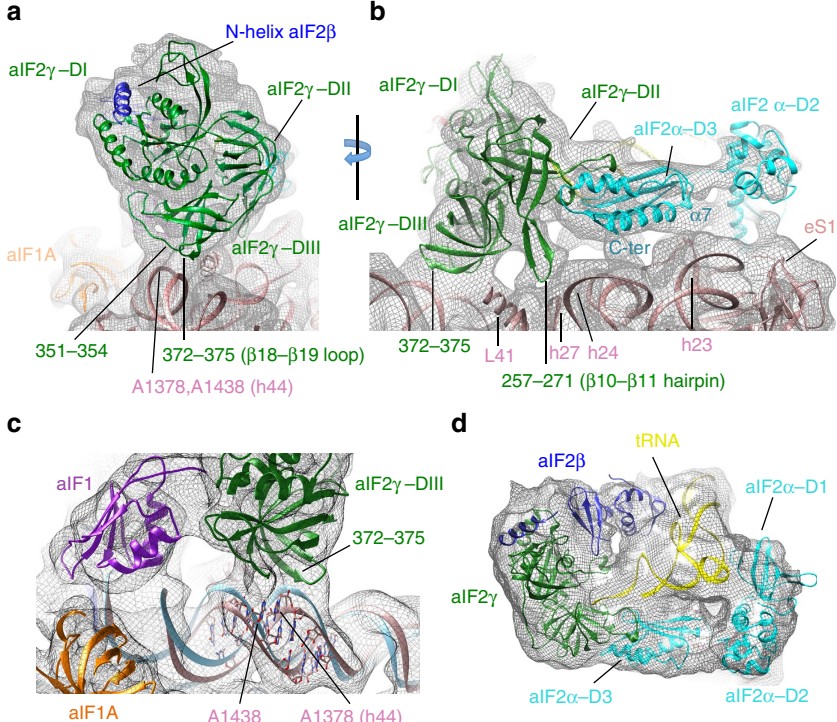

**Figure 4 | Contacts between aIF2 and the 30S ribosomal subunit as observed in IC0–P$_{REMOTE}$.** (**a**) aIF2γ–DIII is in contact with the h44 bulge in 16S RNA. These contacts involve two regions, 351–354 and 372–375, as labelled in green. (**b**) The view is rotated by ∼90° as compared with **a** to highlight the contact of the L2 loop of aIF2γ–DII with h24 (723–725; 769–771 in *E. coli* numbering) and h27 (856–860; 898–902 in *E. coli* numbering), at the hinge between the decoding centre and the platform. Contacts between aIF2α and the ribosome are also visible (see the text for details). (**c**) Fit of h44 (pink) into IC0–P$_{REMOTE}$ map is shown. The view shows that the structure of h44 from Pf-30S subunit (blue, PDB entry 4V6U; ref. 48) had to be remodelled to account for the density. A$_{1378}$ and A$_{1438}$ (Pf numbering) correspond to A$_{1418}$ and A$_{1483}$ in Ec numbering, respectively. (**d**) View showing how aIF2 subunits wrap the initiator tRNA in IC0–P$_{REMOTE}$. The cryo-EM map is low-pass filtered to 8.5 Å.

appears weakened in IC1–P$_{IN}$. In the same view, the two mobile wings of aIF2 (aIF2α–D12 and core of aIF2β) have moved and their fits in the electron density become inaccurate (Fig. 1d and Supplementary Fig. 5). The inside of the L-shaped tRNA, occupied by aIF2β in IC0–P$_{REMOTE}$ is now occupied by aIF1 (Fig. 1). This observation may give sense to the structural homology between aIF1 and aIF2β (refs 26,37). The basic loop of aIF1 moves to allow tRNA accommodation and is now inserted between h44 and the anticodon loop (Supplementary Fig. 5c). The density corresponding to aIF1 appears weaker than in IC0–P$_{REMOTE}$ suggesting reduced interactions in IC1–P$_{IN}$ after start codon recognition. Accordingly, contacts between aIF1 and aIF2 seem looser than in IC0–P$_{REMOTE}$ (Fig. 1c,d). The two IC1–P$_{IN}$ and IC0–P$_{REMOTE}$ structures also differ in the orientation of the 30S head with respect to the body. The observed head movement is similar to that described for py48S when compared with 40S:eIF1:eIF1A (refs 16,22), and corresponds to a clockwise rotation around h28 of about 6° from IC0–P$_{REMOTE}$ to IC1–P$_{IN}$ (Fig. 6a and Supplementary Fig. 7b). By analogy with py48S, the head rotation would allow interaction of G$_{1298}$ and A$_{1299}$ (G$_{1338}$ and A$_{1339}$ in *E. coli* numbering) near h29 with the minor groove G:C base pairs whereas h31 re-positioning would avoid clashes with the anticodon of the tRNA.

## Discussion

Comparison of IC0–P$_{REMOTE}$ and IC1–P$_{IN}$ highlights the structural transition that occurs on start codon recognition. The tRNA accommodation can be described as an isomerization of the IC0–P$_{REMOTE}$ complex whereby the tRNA molecule, surrounded by the α- and β-subunits of aIF2, swings into the P site using the 3′ methionyl-adenosine as a pivot point (Fig. 6 and Supplementary Movie 1). This movement is accompanied by the head rotation allowing locking of the tRNA anticodon stem in the P site. Because the 3′ methionyl-adenosine remains bound to aIF2γ, the establishment of the codon:anticodon interaction induces a structural constraint in the TC. This suggests that aIF2 exerts a spring force on the tRNA adjusted in the P site, which brings back the tRNA in the 'remote' conformation if there is no correct start codon in the mRNA available. On the contrary, recognition of the AUG start codon and formation of correct codon–anticodon pairing, would compensate for the spring force on the tRNA, allowing the tRNA to remain longer in the P site thus triggering further events related to start codon recognition (Fig. 6).

In the P$_{OUT}$ state described for the eukaryotic IC$^{14,22}$, the tRNA is already engaged in the P site. The P$_{REMOTE}$ complex observed here was rather unexpected. Therefore, to further probe the existence of the IC0–P$_{REMOTE}$ conformation, we used the S1 nuclease, known to cleave initiator tRNAs in their anticodon loops$^{38,39}$. Indeed, the P$_{REMOTE}$ state features a tRNA having an anticodon loop expected to be more accessible to nuclease cleavage than that of a tRNA engaged in the P site. We therefore prepared two complexes. The first one corresponds to the complex studied by cryo-EM except that the mRNA contains a CAU codon in place of the AUG start codon, to favour the P$_{REMOTE}$ conformation. The second complex contains the AUG bearing mRNA, but to favour accommodation of the tRNA in the P site, aIF1 was omitted. As shown in Supplementary Fig. 8, the rate of cleavage in the anticodon loop by the S1 nuclease was higher in the former complex than in the latter (2.1 ± 0.07% min$^{-1}$ versus 1.2 ± 0.08% min$^{-1}$). This result is

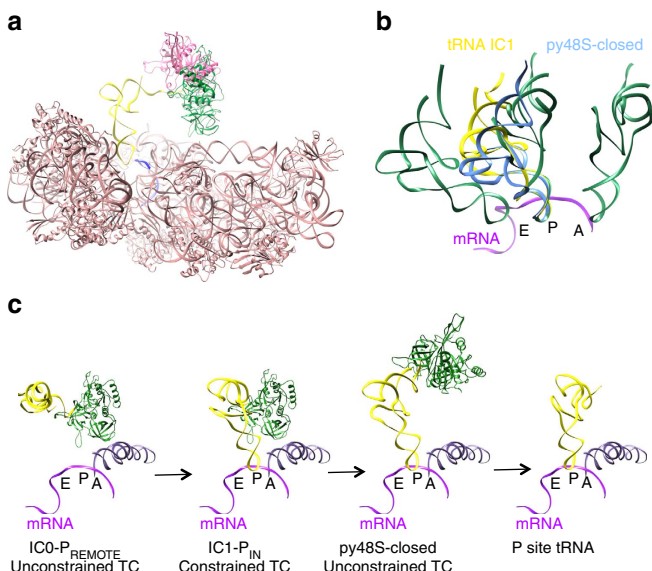

**Figure 5 | Adjustment of the initiator tRNA in the P site.** (**a**) The tRNA in the free TC (PDB entry 3V11) has been superimposed on the tRNA in IC1–$P_{IN}$. aIF2$\gamma$ in free TC is in pink. IC1–$P_{IN}$ is coloured as follows; 30S subunit in light pink, tRNA in yellow, mRNA in blue and aIF2$\gamma$ in green. For the sake of clarity, other components are not represented. The view illustrates that the TC is constrained in IC1–$P_{IN}$. (**b**) Comparison of IC1–$P_{IN}$ tRNA (yellow) with P site tRNAs in 70S (PDB entry 4V51; ref. 34) or in py48S-closed PIC (blue; PDB entry 3JAP; ref. 14). A and E site tRNAs are from PDB entry 4V51, mRNA is from PDB entry 1JG0 (ref. 53). E, P and A tRNAs are coloured in green. (**c**) Cartoon comparing the position of aIF2$\gamma$–tRNA in different ribosomal initiation complexes. h44 (purple) and P site tRNA (yellow) from the 70S *T. thermophilus* complex (PDB entry 4V51; ref. 34) are represented on the right for reference. The mRNA (magenta) is from PDB entry 1JG0 (ref. 53). The relative positions of aIF2$\gamma$–tRNA in IC0–$P_{REMOTE}$, in IC1–$P_{IN}$ and py48S-closed were deduced from superimposition of the body of the small ribosomal subunit of each complex on that of the *T. thermophilus* 70S. The aIF2$\gamma$ or eIF2$\gamma$ are coloured in green. The view illustrates that the TC is constrained in IC1–$P_{IN}$, whereas it is unconstrained in IC0–$P_{REMOTE}$ and py48S-closed complex.

consistent with an increased accessibility of the anticodon loop during the start codon search process, and gives therefore further credence to the existence of the $P_{REMOTE}$ state.

In eukaryotic initiation, recognition of the AUG codon was proposed to cause eIF1 dissociation from the pre-initiation complex, triggering the release of Pi from eIF2 (ref. 7). This model gives eIF1 a crucial gatekeeper role. Consistently, here, the steric constraint imposed by the tRNA pairing at the P site destabilizes aIF1, as shown by its weaker electron density in IC1–$P_{IN}$ compared with IC0–$P_{REMOTE}$ (Fig. 1). Interestingly, the present archaeal structures show contacts between aIF1 and aIF2. One of these contacts involves the switch regions of aIF2, known to control its nucleotide state. This gives clues on how communication between e/aIF1 and e/aIF2 may occur. Indeed, Pi binding on e/aIF2 may be stabilized through the contacts of the switch regions of aIF2$\gamma$ with the N-domain of e/aIF1. On e/aIF1 departure, release of these contacts would destabilize the Pi, leading to its release and to the further dissociation of e/aIF2-GDP. Notably, in addition to switch on/off movements, transition from the GTP to the GDP state of aIF2$\gamma$ is accompanied by a 14° rotation of domains II–III with respect to domain I (ref. 40). It is tempting to speculate that the conformational changes within aIF2$\gamma$ may also facilitate aIF2 departure by favouring the release of the contacts between aIF2$\gamma$–DIII and h44.

The occurrence of the spring force mechanism in archaea raises the question of whether it could be a general mechanism in eukaryotes during the start codon selection process. Interestingly, the rather slow rate of mRNA scanning during CAP-dependent initiation (5–10 nt s$^{-1}$; ref. 41) makes it likely that a constrained state of the TC is also pivotal in eukaryotes. Because constrained states are intrinsically less stable, it is possible that the previously described py48S closed complex[14] represents a later step of the process in which eIF2 has been released from its constrained position on h44 (Fig. 5c). Indeed, in py48S-closed, the TC is unconstrained and no contact between aIF2$\gamma$ and the ribosome is visible. Here, the use of a hyperthermophilic model organism may have helped trap the constrained complex as an important intermediate. Notably, contacts between eIF2 and helix h44, as well as contacts between eIF2 and eIF1, although not observed in the previously determined structures[13,14,22], would make sense in eukaryotic complexes. As a matter of fact, directed hydroxyl radical probing experiments have suggested close vicinity of eIF2$\gamma$–DIII and h44 in yeast[18]. Moreover, biochemical studies have proposed that eIF1 release triggers the release of the Pi group from eIF2 (ref. 7). In this view, it is remarkable that the regions of contact between aIF2$\gamma$ and aIF1, as identified here, are the targets of Sui$^{-}$ mutations in *Saccharomyces cerevisiae* orthologues[8,42,43] suggesting that such interactions may be at the origin of communication between eIF2 and eIF1 during eukaryotic translation initiation. Taken together, we propose that the constrained $P_{IN}$ state observed here represents a new key intermediate during start codon search involving aIF2-assisted codon probing by the initiator tRNA. This provides a novel view of the central role that e/aIF2 plays in start codon selection.

## Methods

**Growth of *P. abyssi* and preparation of ribosomal particles.** *P. abyssi* cells were grown under anaerobic conditions[44]. Mass cultivations were carried out at 90 °C in a 160 l fermenter (Laboratoire de microbiologie industrielle UMR FARE 614, Reims, France). The cells were collected by centrifugation, aliquots were flash-frozen in liquid nitrogen and stored at −80 °C until further use.

To isolate archaeal ribosomes, 3 g of cell pellets were dissolved in 16 ml of buffer A (10 mM Hepes pH 7.5, 100 mM NH$_4$Cl, 10.5 mM magnesium acetate, 0.1 mM EDTA, 6 mM 2-mercaptoethanol, 0.1 mM PMSF, 0.1 mM benzamidine) at 4 °C, mixed with 16 ml of glass beads (5 µm, Sigma) and disrupted by using a Vibrogen-Zellmühle apparatus (Edmund Bühler, Fisher Scientific). After extensive washing of the glass beads and centrifugations, the crude extract (34 ml) was loaded onto a 35 ml sucrose cushion (10 mM Hepes pH 7.5, 1.1 M sucrose, 1 M NH$_4$Cl, 10.5 mM magnesium acetate, 0.1 mM EDTA, 6 mM 2-mercaptoethanol, 0.1 mM PMSF, 0.1 mM benzamidine). After centrifugation at 235,000g for 19 h at 4 °C, the pellet was dissolved in 3 ml of buffer A and loaded onto a Sephacryl S400 column (11 mm × 60 cm, GE Healthcare). The fractions containing ribosomal particles were pooled, dialysed for 3 h against buffer B (same as buffer A but 2.5 mM magnesium acetate) and then loaded on the top of sucrose gradients (10–30% sucrose in buffer B made using BioComp Gradient Master, Fredericton, Canada). After a 19 h centrifugation at 70,000g (SW32.1 rotor, Beckman), the gradients were fractionated. Fractions containing 30S subunits were pooled and the magnesium concentration was increased to 10.5 mM. After centrifugation (220,000g, 20 h), the pellet was briefly rinsed with buffer A to remove residual sucrose and then dissolved in buffer A. About 400 µl of 30S subunits at 2.7 µM were usually obtained. The 30S subunits obtained using this procedure were quite pure, as shown by native and SDS–PAGE (SDS–polyacrylamide gel electrophoresis) analyses (Supplementary Fig. 1b). However, a contaminating band was systematically visible on SDS gels. This band was identified as phosphoenol pyruvate synthase by peptide mass fingerprinting. This high molecular weight protein results from the octameric assembly of a 91 kDa monomer.

**Purification of initiation factors and of aminoacyl-tRNAs.** Pa-aIF2—Cultures (250 ml) of cells overproducing each of the three subunits (native aIF2$\gamma$, N-terminally tagged versions of aIF2$\alpha$ and aIF2$\beta$) were harvested, mixed in 40 ml of buffer C (500 mM NaCl, 10 mM HEPES pH 7.5, 10 mM 2-mercaptoethanol, 0.1 mM EDTA, 0.1 mM PMSF, 0.1 mM benzamidine) and disrupted by sonication. After centrifugation, the supernatant was heated for 10 min at 80 °C. After pelleting the nonthermostable proteins by centrifugation, the supernatant was directly loaded onto a Q-Hiload column (10 mm × 4 cm; GE Healthcare) equilibrated in buffer C. The flow-through was recovered and diluted twofold with buffer D (10 mM HEPES pH 7.5, 10 mM 2-mercaptoethanol, 0.1 mM EDTA, 0.1 mM PMSF,

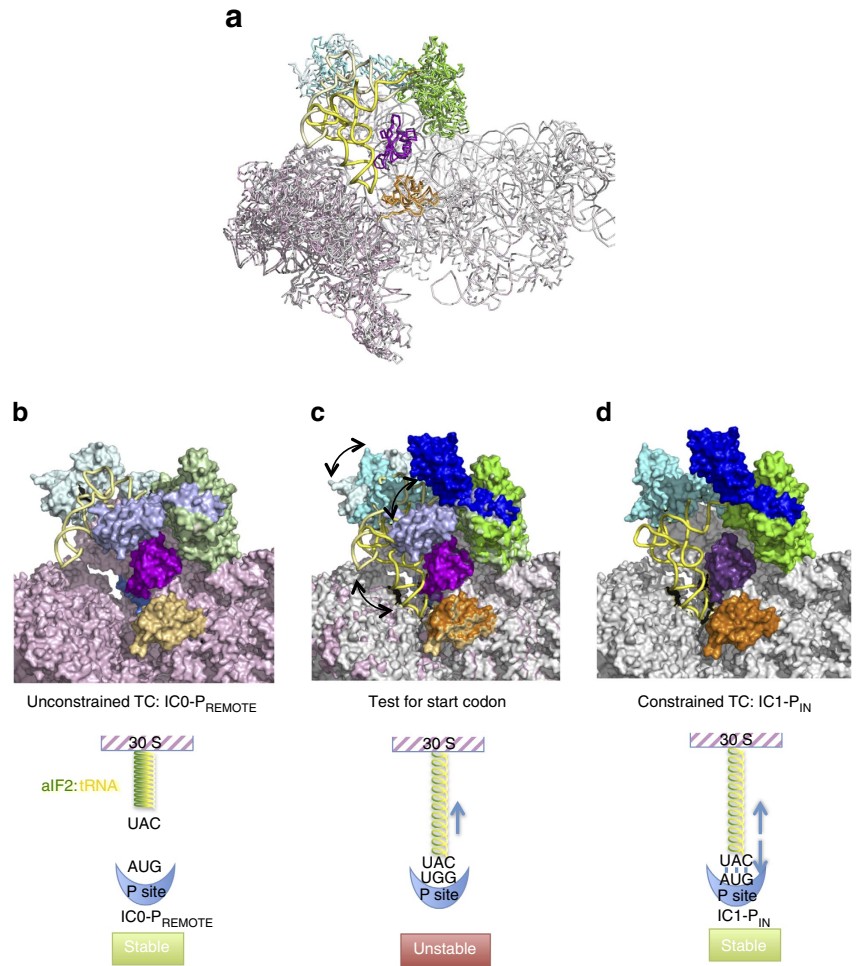

**Figure 6 | Model of the role of aIF2 in start codon selection. (a)** The positions of the TC in IC0–$P_{REMOTE}$ and IC1–$P_{IN}$ are compared after superimposition of the 30S bodies. Colour code for IC0–$P_{REMOTE}$ is as follows: ribosome in pink, tRNA in light yellow, aIF2γ in light green, aIF2α in light cyan, aIF1 in light pink, aIF1A in light orange and mRNA in blue. Colour code for IC1–$P_{IN}$ is as follows: 30S head in grey, tRNA in yellow, aIF2γ in green, aIF2α in cyan, aIF1 in magenta, aIF1A in orange and mRNA in blue. For the sake of clarity, aIF2β is not represented. The view shows that aIF2γ does not significantly move during the IC0–$P_{REMOTE}$ to IC1–$P_{IN}$ transition. Therefore, the TC is distorted in IC1–$P_{IN}$. The view also shows that accommodation of the tRNA in the P site is accompanied by a rotation of the 30S head. **(b)** Surface representation of IC0–$P_{REMOTE}$: colour code is same as in **a** with aIF2β in light blue. The TC has the same conformation as unbound to 30S and is therefore unconstrained. **(c)** Surface representation of IC0–$P_{REMOTE}$ superimposed on IC1–$P_{IN}$. The tRNA can explore the P site although this results in a constrained TC conformation. **(d)** Surface representation of IC1–$P_{IN}$. Colour code is same as in (**a**) with aIF2β in dark blue. If an AUG codon is found in the P site, the constrained conformation of the TC is stabilized by base pairing. The lower part of the figure shows schemes illustrating the spring force model.

0.1 mM benzamidine). The sample was then loaded onto an S-Hiload column (10 mm × 4 cm; GE Healthcare) equilibrated in buffer D containing 250 mM NaCl. Assembled heterotrimer was eluted by applying a step of buffer D containing 700 mM NaCl. The recovered sample was then concentrated to 1 ml. The final heterotrimer preparation was obtained after purification by molecular sieving on a Superdex 200 HR column (10 mm × 30 cm, GE Healthcare). At the end of purification, the protein was concentrated and 1 mM Gpp(NH)p-$Mg^{2+}$ was added. Flash-frozen samples were stored at − 80 °C.

Pa-aIF1—The gene encoding aIF1 from *P. abyssi* was amplified from genomic DNA and cloned into pET15blpa to produce an N-terminally his-tagged version of the factor. A culture (250 ml) of cells overproducing the tagged version of aIF1 was collected, re-suspended in 10 ml of buffer D plus 100 mM NaCl and heated for 10 min at 70 °C. After pelleting non-thermostable proteins by centrifugation, the supernatant was directly loaded onto a Talon affinity column (10 mm × 4 cm; Clontech) equilibrated in buffer D plus 100 mM NaCl. The tagged protein was eluted with a step of buffer D containing 125 mM imidazole. After dialysis against buffer D containing 250 mM NaCl, the preparation was passed through a Q-Hiload column equilibrated in buffer A containing 100 mM NaCl. This step was repeated twice to efficiently remove nucleic acids. After concentration, the protein was flash-frozen and stored at − 80 °C. Routinely, 4 mg of purified protein were obtained from a 250 ml culture.

Pa-aIF1A—The gene encoding aIF1A from *P. abyssi* was amplified from genomic DNA and cloned into pET3alpa to produce a native version of the factor. A culture (250 ml) of cells overproducing the native version of aIF1A was collected,

resuspended in 10 ml buffer A containing 500 mM NaCl and disrupted by sonication. After centrifugation, the supernatant was heated for 10 min at 80 °C. After pelleting the non-thermostable proteins by centrifugation, the supernatant was directly loaded onto a Q-Hiload column (10 mm × 4 cm; GE Healthcare) equilibrated in buffer A plus 500 mM NaCl. The flow-through was recovered, diluted twice to obtain a final concentration of 250 mM NaCl and loaded onto an S-Hiload column (10 mm × 4 cm; GE Healthcare) equilibrated in buffer A containing 250 mM NaCl. aIF1A was eluted by applying an NaCl gradient (250 to 800 mM) in buffer A. The recovered sample was then concentrated, flash-frozen and stored at − 80 °C. Routinely, 4 mg purified protein were obtained from a 250 ml culture.

**Preparations of methionylated tRNAs.** $tRNA_f^{Met}$ $A_1$-$U_{72}$ was produced in *E. coli* from a cloned gene, purified and aminoacylated as described[45]. As previously shown, $tRNA_f^{Met}$ $A_1$-$U_{72}$ from *E. coli* is 79% identical to Pa-$tRNA_i^{Met}$ and the two methionyl-tRNAs bind Pa-aIF2 with almost the same affinity[28]. Moreover, SAXS data have shown that an aIF2–tRNA complex formed with either Met-$tRNA_f^{Met}$ from *E. coli* or archaeal Met-$tRNA_i^{Met}$ display identical diffusion curves, which strongly suggest similar structures[31].

**Purification of IC.** The IC was assembled as follows: 864 picomoles of Pa-30S were mixed to a twofold excess of Pa-aIF2 (1,728 picomoles), fivefold excess of Pa-aIF1 (4,322 picomoles), Pa-aIF1A, met-$tRNA_f^{Met}A_1$-$U_{72}$, synthetic mRNA ($A_{(-17)}$

UUU**GGAGGUGA**UUUAAA$_{(+1)}$**UG**CCAAAG$_{(+9)}$, ThermoScientific) in the presence of GDPNP to a final concentration of 1 mM. The sequence of this synthetic mRNA corresponds to the natural start region of the mRNA encoding elongation factor aEF1A from *P. abyssi*. Note that the ternary complex was preformed first by mixing aIF2, GDPNP, Met-tRNA$_f$$^{Met}$A$_1$-U$_{72}$ and incubating 30 s at 51 °C. The IC mixture was then incubated for 5 min at 51 °C in IC buffer (10 mM Hepes pH 7.5, 100 mM NH$_4$Cl, 10 mM magnesium acetate, 3 mM 2-mercaptoethanol) and 1 h 45 min at room temperature. The IC complex was purified using Talon affinity column and standard procedures. The recovered complex (Supplementary Fig. 1) was dialysed against IC buffer and concentrated to 2.6 μM using centricon 30, and then stored at −80 °C after flash freezing in liquid nitrogen. The presence of all the components was confirmed by SDS–PAGE (Supplementary Fig. 1b).

**S1 nuclease as a probe of tRNA anticodon loop accessibility.** Two ICs were assembled as described above except that we used an excess of all the components over initiator tRNA to ensure its full binding to the 30S subunits. The first complex contained 0.5 μM Pa-30S, 1 μM of synthetic mRNA identical to that used for Cryo-EM studies but containing a CAU codon instead of the AUG start codon (CAU mRNA), 1 μM aIF2, 50 μM GDPNP, 8 mM MgCl$_2$, 0.37 μM Met-tRNA$_f$$^{Met}$A$_1$U$_{72}$, 1 μM aIF1A and 2 μM aIF1. The second complex was identical except that the standard synthetic mRNA with AUG was used and that aIF1 was omitted. After assembly (8 μl samples) and incubation for 5 min at 51 °C, 1 μl of 10× S1-buffer (300 mM MOPS pH 6.7, 10 mM zinc acetate, 50% glycerol) and 1 μl of S1 nuclease (four units, NEN, from laboratory stocks) were added. S1 nuclease hydrolysis was performed at 51 °C during various incubation times. The reaction was stopped by adding 2 μl of stop buffer (50% formamide, 2 mM ATP, 10 mM EDTA, 0.03% xylene cyanol, 0.03% bromophenol blue). The samples were heated for 3 min at 95 °C before loading onto a 12% polyacrylamide gel containing 50% urea. The gel was stained with ethidium bromide and then analysed with Image J (ref. 46). As previously shown[38,39], S1 nuclease cleaves in the anticodon loop. The percentage of tRNA cleavage was plotted as a function of time and the rate of cleavage deduced from linear regression (Supplementary Fig. 8).

**Electron microscopy.** Before spotting onto the grids, an excess of mRNA, aIF1A, TC (each 1:1 regarding IC) and GDPNP (0.5 mM) was added and the IC was heated 1 min at 50 °C. After heating, the samples were diluted to a final concentration of ∼0.2 μM in IC buffer containing 0.5 mM GDPNP and immediately used for grid preparations. Carbon-coated Quantifoil R2/2 grids (Quantifoil Micro Tools GmbH) were blotted with 4 μl of the purified samples and vitrified in liquid ethane at 100 K on a Vitrobot Mark IV (FEI). Low-dose images were collected at a calibrated magnification of ×129,629 on an FEI Titan Krios with a Cs corrector operating at 300 kV, using a 4 × 4k FEI Falcon II camera (Netherlands Centre for Electron Nanoscopy, Leiden). The pixel size was 1.12 Å on the camera. De-focus values ranged from −0.5 to −2.8 μm. All the images were recorded using the EPU software (FEI) for automatic data collection.

**Image processing.** The low-dose images were taken with a total dose of 44 e$^-$ Å$^{-2}$ summed on 16 frames. The first frame and the last eight frames were discarded to minimize beam-induced movement and irradiation. The remaining frames (corresponding to a total dose of 19 e$^-$ Å$^{-2}$) were first aligned with DRIFTCORR[47], averaged and then sorted according to best power spectrum and ice quality. The particles from the selected images (2,154) were automatically picked with the RELION package[21]. An initial data set of ∼192,000 particles was sorted using several rounds of classical two-dimensional classification protocols implemented in RELION. Then, a 3D-classification in RELION using a 70 Å filtered map of the archaeal 30S (extracted from the *Pyrococcus furiosus* 70S structure[48]) helped to isolate several well-resolved classes that were used for further 3D classification. The following sorting procedure allowed us to discard classes without or with poorly resolved aIF1, aIF1A or tRNA. Some of the remaining classes still represented a mixture of conformations (IC0–P$_{REMOTE}$ and IC1–P$_{IN}$) and needed further 3D-classification. Finally, well-resolved classes representing unambiguous positions of the tRNA were regrouped and used for individual 3D refinement (Supplementary Fig. 2a). Two final data sets of ∼12,600 and ∼5,500 particles were used to get the final 3D reconstructions called IC0–P$_{REMOTE}$ and IC1–P$_{IN}$, respectively. The resolutions of the final 3D reconstructions were estimated, using gold-standard Fourier Shell Correlation, to 9.5 and 12.6 Å for IC0–P$_{REMOTE}$ and IC1–P$_{IN}$, respectively. A movie processing approach[49] was also attempted in RELION but did neither increase the maps quality nor their resolution. Finally, a post-processing approach also implemented in RELION gave higher resolution maps of 5.34 and 7.5 Å (Supplementary Fig. 2b) for IC0–P$_{REMOTE}$ and IC1–P$_{IN}$, respectively. Reported overall resolutions were calculated with the gold-standard 0.143 FSC cut-off criterion[21] and were corrected for the effects of a soft mask on the FSC curve with high-resolution noise substitution[50]. The final density maps were corrected for the modulation transfer function of the detector and sharpened by applying a negative B factor that was estimated using automated procedures[51]. Sharpening B-values were −132 Å$^2$ for IC0–P$_{REMOTE}$ and −208 Å$^2$ for IC1–P$_{IN}$.

To assess the quality of the final reconstructions, a local resolution map for each reconstruction was also computed: information up to 4 Å, for the core of the 30S subunit and between 6 and 15 Å for the translation initiation factors binding regions is available in both complexes (Fig. 2 and Table 1). Such decrease in resolution as distance with the core 30S increases was already observed in other ICs[22]. Because of this, for better clarity in the figures showing factors, maps were generally low-pass filtered to 8.5 Å (IC0–P$_{REMOTE}$) or 12 Å (IC1–P$_{IN}$).

**Model building and refinement.** *IC0–P$_{REMOTE}$.* The structures of the head and body parts of the 30S subunits were extracted from the *P. furiosus* 70S coordinates[48] (PDB entry 4V6U). The head and body were manually placed independently into the electron density map using Chimera, before automatically refining the position in the same programme. Messenger RNA was clearly visible in the density, including interaction of the SD sequence with the anti-SD (Supplementary Fig. 4a). The mRNA was modelled in Coot[52] using the structure of the *T. thermophilus* 70S ribosome complexed with mRNA and tRNAs (PDB entry 1JGO[53]) as a guide. The start codon was positioned in the P site. The mRNA was not visible from base +5 to base +9 (as in yeast 48S complex, PDB entry 3JAQ) at the 3′ extremity. Bases −17 and −16 at the 5′ end of mRNA were not visible. The isolated PDB models of initiation factors and tRNA were manually placed in the electron density. The crystallographic model of the ternary IC aIF2:GDPNP:tRNA[23] (PDB entry 3V11) could be readily fitted in the electron density with few modifications (see text, Fig. 4d and Supplementary Fig. 3b). Finally, a few manual model corrections were performed in Coot[52] to better fit the electron density maps. These corrections concerned the following regions: 16S rRNA connection between the head and body, h44 bulge, aIF1 basic loop, h44 near aIF1A. The obtained model was then refined in Refmac[54] using data up to 5.8 Å resolution. Tight geometric and external restraints (generated using ProSmart and LIBL) were maintained throughout refinement to minimize overfitting, as described[55–57]. According to the relatively modest resolution, an overall B value was used for the whole model. Average Fourier shell correlation (FSC) was monitored in Refmac. For cross-validation against overfitting, the atoms of the final model were randomly displaced (with a maximun deviation of 0.5 Å), and a refinement procedure was performed against the maps that were reconstructed from only one of the two independent halves of the data used in our gold-standard FSC procedure. The FSC curves were calculated between the resulting model and the half map against which it had been refined (FSCwork) as well as the FSC curve between the model and the other half map (FSCtest)[55] (Supplementary Fig. 2c). The average FSC for the final model against the 5.8 Å resolution data used in refinement was 0.56 (Rfactor 0.42). The correlation coefficient between the map simulated from the model (5.34 Å resolution) and the experimental map was 82.1%, as determined using Chimera[58]. Further attempts to improve this value by loosening restraints during refinement systematically led to overfitting. Therefore, at the present resolution, Refmac was essentially used to fix the small clashes and to improve geometry of the initial 30S unrefined cryo-EM model. The number of bad bonds dropped from 5.93 to 0.06%, and the number of bad angles from 6.93 to 0.58%. The statistics for image processing and refinement are summarized in Supplementary Table 1.

*IC1–P$_{IN}$.* The refined head, body and mRNA from IC0–P$_{REMOTE}$ were fitted within IC1 map using Chimera. The positions of aIF1 and aIF1A were manually adjusted in the electron density. According to its known structural organization[25,26], aIF2 was placed as three rigid bodies: (1) the γ-subunit bound to domain 3 of the α-subunit and to the N-terminal helix of the β-subunit, (2) domains 1 and 2 of the α-subunit and (3) the core domain of the β-subunit. The model of the tRNA was built using the structure of the P site initiator tRNA as observed in 4V51 (ref. 34). The tRNA was cut into two parts in the hinge region (G$_{26}$ A$_{44}$). The two parts were fitted independently. Start codon and anticodon tRNA bases were adjusted to allow base pairing. Moreover, the acceptor CCA end was remodelled to allow fitting of the methionylated 3′adenosine in its pocket on the aIF2γ–DII domain. Finally, the connection between the two tRNA halves was manually adjusted. As a final step, the basic loop of aIF1 was rebuilt to better fit the density. Because of the modest resolution of IC1–P$_{IN}$, Refmac refinement systematically led to overfitting and was therefore not applied to the final model. Correlation coefficient between the map simulated from the model (7.5 Å resolution) and the experimental map was 88.0%, as determined using Chimera[58].

**Data availability.** The EMDataBank accession numbers for the EM maps reported in this paper are EMD-8148 (IC0–P$_{REMOTE}$) and EMD-8149 (IC1–P$_{IN}$). The coordinates of the models fitted in the maps have been deposited in the Protein Data Bank (accession numbers 5JB3 for IC0–P$_{REMOTE}$, 5JBH for IC1–P$_{IN}$). The data that support the findings of this study are available from the corresponding author on request.

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

## Acknowledgements

This paper is dedicated to the memory of our friend and colleague Nicolas Boisset. We thank F. Weis, R. Gillet, S. Jonic, A. Myasnikov for valuable discussions on electron microscopy and J.F. Menetret for help with data collection at the preliminary stages of this study. We also gratefully acknowledge R. Matadeen and S. de Carlo (FEI Company)

for image acquisition at the National Center for Electron Nanoscopy in Leiden (NeCEN) and helpful advice. We thank Thomas Simonson for critical reading of the manuscript. This work was supported by the Centre National de la Recherche Scientifique, Ecole polytechnique, the Physique et Chimie du Vivant program of Agence Nationale de la Recherche (MASTIC project) and by the French Infrastructure For Integrated Structural Biology (FRISBI) ANR-10-INSB-05-01. A.M. was supported by a 'Bourse Monge' doctoral fellowship from Ecole polytechnique.

## Author contributions

P.-D.C., E.S and Y.M. designed the research; P.-D.C., Y.M., E.S., C.L.-S., A.M., E.L. and L.C. carried out the experiments; all the authors analysed the data; P.-D.C., E.S., Y.M. and B.P.K. wrote the paper.

## Additional information

**Competing financial interests:** The authors declare no competing financial interests.

**Publisher's note**: 

