## [Peer Review File · Nature Communications]

Reviewers' comments:

Reviewer #1 (Remarks to the Author):

Translation initiation is the rate-limiting step during protein synthesis and is catalyzed by specialized translation initiation factors. While a number of structures have been reported of 30S translation initiation complexes from eukaryotes and bacteria, so far no structures are available from archaea. Archaeal translation initiation can be thought of as a simplified version of eukaryotic initiation since it requires only three initiation factors (rather than 13 in eukaryotes), namely aIF1, aIF1A and aIF2. Analysis of archaeal translation initiation complexes can therefore provide not only insight into archaeal translation but may also be in some ways mechanistically similar to eukaryotic initiation.

The manuscript of Coureux and co-workers reports two cryo-EM structures of archaeal 30S translation initiation complexes, namely, a state with the initiator tRNA engaged in codon-anticodon interactions and a second where the anticodon stem loop of the initiator tRNA does not contact the 30S subunit. Overall, the manuscript is well-presented however the authors need to make clearer the differences and similarities with previous eukaryotic initiation complexes as well as highlight the novel findings to emphasize better the mechanistic advance compared to previous structures.

Major criticisms:

1. Page 1, the title does not accurately reflect the content of the research. At a minimum it should reflect that this is a structural paper on archaeal 30S initiation complexes.
2. The authors need to either tone down their adjectives or provide a better basis for them. For example, page 2 "two unprecedented conformational states" is rather exaggerated since the tRNA "IC1-PIN" conformational is more or less identical those observed previously for eukaryotic initiation.
3. As mentioned above, the authors highlight in many places how studying archaeal translation initiation complexes can provide insight into eukaryotic initiation, for example, on page 3, lines 66-74, however, it is surprising that there are no figures showing the direct structural comparisons of all the different complexes from the Frank and Ramakrishnan groups? The only comparisons are in Sup Fig. 5 but only with respect to the head movement of the 30S subunit.
4. As mentioned above, the authors need to highlight better the advance in mechanistic insights beyond the available eukaryotic structures. The eIF1 departure triggering eIF2 Pi release is not clearly explained and appears to be simply speculation based on contact. On page 9, line 222 it is written as a fact.

5. The use of the term IC0-POUT is rather confusing since there is already a previous reported POUT state, which is quite different to the one reported here since contact of the tRNA with the 30S was still maintained. I understand that the state reported here is even more OUT than the previous one, but giving it a similar name does not distinguish it sufficiently.
9. Figure 2 needs a complementary supplementary figure comparing archaeal and previous eukaryotic states. In fact, I am very surprised that this was not included in the original manuscript since this is important to highlight similarities between this structure and previous ones. The reader needs to understand what is novel and how this leads to new insights!
10. Technically, the reconstructions are satisfactorily. It is worth mentioning that reconstructions of small subunit complexes are more difficult than 70S/80S ribosomes and therefore the resolutions of 5.3 and 7.5 reported here is acceptable and comparable with previous small subunit reconstructions. However, it is also important that the authors make it clearer earlier in the manuscript that although the overall resolution is 5.3 and 7.5, the ligands are much worse resolved and in fact the maps are filtered to 8.5 and 12A for most figures. Currently it is not till page 4 that the reader can find the resolutions and then only the overall resolutions which are not particularly relevant for the ligands. I find this quite misleading and therefore needs to be made clearer earlier!

Minor criticisms:

1. Page 5, line 127. It is not clear why this interaction is "striking" - perhaps the authors should leave this subjective viewpoint to the reader to decide.
2. Page 5, line 130. The authors use a numbering that is unclear. I presume archaeal? It would make sense to at least include E.coli numbering as well, especially when making reference to decoding nucleotides such as A1492 and A1493 (e.g. page 7, line 168).
3. Page 6, line 156: This sentence needs to be rephrased. This is the first archaeal structure so its hard to understand how aIF1 can occupy its previously described position. Presumably the authors are referring to the position of eIF1?
4. There are endless descriptive details of interactions in this paper. However, it is not clear which ones were observed in the eukaryotic complexes and which ones are novel?
5. Page 7, line 167, aIF1A was not present in the eukaryotic PIC!!
6. Page 7, lines 186-187: This comparison could be included in a supplementary figure.
7. The authors might get a native English person to go through the manuscript. There are a number of improvements that could be made: For example, Page 8, line 193: perhaps "inaccurate" is a more appropriate term than "poorly accurate"? Page 8, line 197: perhaps "weaker" is a more appropriate term than "less strong"? Page 8, line 198: perhaps "looser" is a more appropriate term than "less tight"? etc etc...
8. It would be beneficial to have Figure 1 and 2 combined with four panels and using the same angles so that the views in figure 2 can be related to the overviews in Figure 1.
9. Page 18, aIF1A mRNA? Is this the name of the mRNA, or is this mRNA encoding aIF1A? Perhaps the authors should consider renaming the mRNA if it's the former to avoid confusion.

10. Page 19, 1.75h - does this mean 1h 45min?
11. Page 19, the authors should provide a total dose for frames used in the final reconstructions.
12. Page 20: From 192000 particles only 12600 and 5500 remained in the final reconstructions. This is seen in Figure 1d, however, the figure is simply too small to see what all these different states are and how they differ. It would be appropriate to describe the different states in some details in the methods.
13. The statistics for processing and refinement are hidden in the methods. It would be easier for the reader if the authors simply made a supplementary table to summarize all the numbers, like most other structural papers.

Reviewer #2 (Remarks to the Author):

Selection of the AUG start codon during translation initiation in eukaryotic cells requires the base-by-base scanning of the mRNA leader until complementary base-pairing interactions are established between the anticodon loop of the initiator Met-tRNA_i and the start codon in the mRNA. Genetic, biochemical and structural insights have led to a model whereby the Met-tRNA_i in complex with eIF2 binds to the P site of the 40S ribosomal subunit in a P(out) conformation where it can inspect the mRNA sequence. Upon codon-anticodon pairing, the tRNA moves to a P(in) conformation and the 40S subunit adopts a closed conformation around the mRNA that blocks further scanning. Structural support for this model has come most recently from high-resolution (4-6 Å) cryo-EM structures of preinitiation complexes from yeast.

In this paper the authors provide high-resolution (5.3 and 7.5 Å) structures of preinitiation complexes from archaea. While archaeal translation shares some resemblance with eukaryotic translation with the common requirement for translation initiation factors eIF1, eIF1A and eIF2, eukaryotic translation requires additional factors including the GTPase activating protein eIF5. Moreover, selection of the start codon during archaeal translation is facilitated by a bacterial-like Shine-Dalgarno interaction between mRNA sequences preceding the start codon and sequences near the 3' end of the rRNA in the small ribosomal subunit.

In the lower resolution structure in this paper, the Met-tRNA_i is in a P(in) conformation, base-paired to the start codon. This structure resembles the structure of the yeast P(in) complex reported previously. However, in contrast to the yeast structure, the eIF2 γ subunit docks on helix h44, consistent with published chemical probing studies. Unexpectedly, the N-terminal domains of eIF2 α and the C-terminal domain of eIF2 β appear to disengage from the core of the structure. This apparent disengagement of eIF2 α is particularly striking as in the yeast structure this domain contacts the context nucleotides flanking the start codon. It is unclear whether these structural differences reflect different states of the eukaryotic and archaeal

complexes or mechanistic differences in the mode of translation initiation between eukaryotes and archaea. Notably, the structure also visualizes the base-pairing between the rRNA and Shine-Dalgarno mRNA sequences.

The higher resolution structure is quite surprising. In this structure the anticodon loop of the tRNA_i is pivoted out of the P site and is remote from the mRNA. While this structure would be incompatible with a scanning mechanism of start codon selection, it could reflect the alternative Shine-Dalgarno mechanism used in archaea. While the authors refer to this structure as P(out), this could be confusing to readers in the translation field as this P(out) does not resemble the P(out) conformation observed in the yeast preinitiation complex structures.

While the paper is written clearly and the structures apparently justify the conclusions, it would be reassuring to have independent evidence to support the unexpected orientation of the Met-tRNA_i in the so-called P(out) structure as well as evidence to support the displacement of eIF2 α and eIF2 β in the P(in) structure.

Other comments:

- 1) line 55: "loses" instead of "looses"
- 2) line 100: why is excess mRNA, aIF1A and TC added to purified complexes before spotting on grids? Could the unexpected "P(out)" conformation reflect a non-functional complex?
- 3) line 134: it might be useful to cite figure 3b here
- 4) Figure 4 legend, line 291: "b-" should be "(c)"
- 5) Supplementary Figure 1a legend: the last word "(right)" is confusing.

Point-by-point responses to the reviews

Former title: Initiator tRNA accommodation during translation initiation mRNA probing

New title : Cryo-EM study of start codon selection during archaeal translation initiation

Pierre-Damien Coureux, Christine Lazennec-Schurdevin, Auriane Monestier, Eric Larquet, Lionel

Cladière, Bruno P Klaholz, Emmanuelle Schmitt and Yves Mechulam

Correspondence : emmanuelle.schmitt@polytechnique.edu

Editor's comments

Your manuscript entitled "Initiator tRNA accommodation during translation initiation mRNA probing" has now been seen by 2 referees, whose comments are appended below. You will see that, while they find your work of interest, they have raised points that need to be addressed before we can make a decision on publication.

We therefore invite you to revise and resubmit your manuscript, taking into account the points raised. In particular, it would be important to provide additional biochemical data to support the structural observation and to put the finding in a better context in regard the advance provided and differences with the eukaryotic systems. At the same time, we ask that you ensure your manuscript complies with our editorial policies.

We thank the editor for offering us the opportunity to submit a revised manuscript. We thank the reviewers for their positive and meticulous assessments of our work. Their detailed suggestions on structure analyses and data presentation were very helpful. Altogether, this has allowed us to substantially revise both text and figures of our manuscript, to include a biochemical experiment giving support to our structural observations and to better put the findings in the context of the recent advances with the eukaryotic systems.

We provide below our point-by-point responses to the referees' comments, which detail our changes to the manuscript. To provide a quick overview of our revisions, we have highlighted in red all new or substantially altered portions of text in the revised manuscript files.

Reviewer #1:

Translation initiation is the rate-limiting step during protein synthesis and is catalyzed by specialized translation initiation factors. While a number of structures have been reported of 30S translation initiation complexes from eukaryotes and bacteria, so far no structures are available from archaea. Archaeal translation initiation can be thought of as a simplified version of eukaryotic initiation since it requires only three initiation factors (rather than 13 in eukaryotes), namely aIF1, aIF1A and aIF2. Analysis of archaeal translation initiation complexes can therefore provide not only insight into archaeal translation but may also be in some ways mechanistically similar to eukaryotic initiation.

The manuscript of Coureux and co-workers reports two cryo-EM structures of archaeal 30S translation initiation complexes, namely, a state with the initiator tRNA engaged in codon-anticodon interactions

and a second where the anticodon stem loop of the initiator tRNA does not contact the 30S subunit. Overall, the manuscript is well-presented however the authors need to make clearer the differences and similarities with previous eukaryotic initiation complexes as well as highlight the novel findings to emphasize better the mechanistic advance compared to previous structures.

Major criticisms:

1. Page 1, the title does not accurately reflect the content of the research. At a minimum it should reflect that this is a structural paper on archaeal 30S initiation complexes.

The title has been modified according to the reviewer's comment. We agree that this allows the reader to get a more accurate idea of the content.

2. The authors need to either tone down their adjectives or provide a better basis for them. For example, page 2 "two unprecedented conformational states" is rather exaggerated since the tRNA "IC1-PIN" conformational is more or less identical those observed previously for eukaryotic initiation.

Thanks to the reviewer comment (see also point 3) a particular attention has been paid to clarifying the differences and the similarities between the present archaeal initiation complexes and the previously determined structures of eukaryotic translation initiation complexes. Moreover, several emphatic statements have been removed throughout the text. (line 31, line 76, line 135, line 279).

3. As mentioned above, the authors highlight in many places how studying archaeal translation initiation complexes can provide insight into eukaryotic initiation, for example, on page 3, lines 66-74, however, it is surprising that there are no figures showing the direct structural comparisons of all the different complexes from the Frank and Ramakrishnan groups? The only comparisons are in Sup Fig. 5 but only with respect to the head movement of the 30S subunit.

Indeed, we agree with the reviewer that our findings were not correctly put in the context of the eukaryotic cryo-EM studies. To answer this point we first added a new figure (Figure 3 in the revised version) showing a comparison of the two states of the archaeal translation initiation complexes with the P_{OUT} and P_{IN} states previously observed using eukaryotic systems. Moreover, throughout the text, particular care was given to better distinguish the features in common with the previous eukaryotic structures from the features that are novel in the present study (e.g. page 5, lines 118-120; page 7, lines 168-169; page 7, line 168 and 172-174).

4. As mentioned above, the authors need to highlight better the advance in mechanistic insights beyond the available eukaryotic structures. The eIF1 departure triggering eIF2 Pi release is not clearly explained and appears to be simply speculation based on contact. On page 9, line 222 it is written as a fact.

According to the reviewer's suggestion, a paragraph has been rewritten in the Discussion section to better explain the current mechanistic model in eukaryotes and to better highlight the new insights from the archaeal structures (page 10 lines 250-262).

5. The use of the term IC0-POUT is rather confusing since there is already a previous reported POUT state, which is quite different to the one reported here since contact of the tRNA with the 30S was still maintained. I understand that the state reported here is even more OUT than the previous one, but giving it a similar name does not distinguish it sufficiently.

We agree with both reviewers that the term IC0-P_{OUT} was inappropriate and confusing. We have therefore called the corresponding complex IC0-P_{REMOTE}.

9. Figure 2 needs a complementary supplementary figure comparing archaeal and previous eukaryotic

states. In fact, I am very surprised that this was not included in the original manuscript since this is important to highlight similarities between this structure and previous ones. The reader needs to understand what is novel and how this leads to new insights!

A figure (Figure 3) has been added in order to illustrate this comparison. See also response to point 3 above.

10. Technically, the reconstructions are satisfactorily. It is worth mentioning that reconstructions of small subunit complexes are more difficult than 70S/80S ribosomes and therefore the resolutions of 5.3 and 7.5 reported here is acceptable and comparable with previous small subunit reconstructions. However, it is also important that the authors make it clearer earlier in the manuscript that although the overall resolution is 5.3 and 7.5, the ligands are much worse resolved and in fact the maps are filtered to 8.5 and 12A for most figures. Currently it is not till page 4 that the reader can find the resolutions and then only the overall resolutions which are not particularly relevant for the ligands. I find this quite misleading and therefore needs to be made clearer earlier!

We thank the reviewer for this remark. Overall resolutions of the two Cryo-EM structures are now stated in the introduction section page 3, together with a sentence stating that the resolution is worse for the factors. Moreover, in order that the reader can find the complete information regarding the resolution of the two structures, local resolution maps and explicit resolution for each factor are now shown in figure 2 and in Table 1 (instead of in a Supplementary figure). Figure 1 and 2 are called in the manuscript in the first paragraph of the results section page 4. To comply with the journal requirements, the figures could not be called in the introduction section.

Minor criticisms:

1. Page 5, line 127. It is not clear why this interaction is "striking" - perhaps the authors should leave this subjective viewpoint to the reader to decide.

The subjective word has been removed.

2. Page 5, line 130. The authors use a numbering that is unclear. I presume archaeal? It would make sense to at least include E.coli numbering as well, especially when making reference to decoding nucleotides such as A1492 and A1493 (e.g. page 7, line 168).

It is now stated in the text that the numbering used for the 16S RNA corresponds to that of *Pyrococcus furiosus* sequence. Moreover, the *E. coli* numbering is also systematically given in parentheses as recommended by the reviewer.

3. Page 6, line 156: This sentence needs to be rephrased. This is the first archaeal structure so its hard to understand how aIF1 can occupy its previously described position. Presumably the authors are referring to the position of eIF1?

The sentence has been rephrased to state that the observed position of aIF1 corresponds to that of eIF1 in eukaryotic complexes. We apologize for this inappropriate shortcut. Other similar shortcuts have also been corrected throughout the text.

4. There are endless descriptive details of interactions in this paper. However, it is not clear which ones were observed in the eukaryotic complexes and which ones are novel?

We agree with the reviewer that some descriptive details in the main text were somewhat heavy for the reader. We have therefore simplified the descriptions (lines 136-144) and mentioned the details such as residue numbers in the figure legends.

5. Page 7, line 167, aIF1A was not present in the eukaryotic PIC!!

The sentence has been rephrased to state that the observed position of aIF1A corresponds to that of eIF1A in eukaryotic complexes (line 180). Again, we apologize for this inappropriate shortcut.

6. Page 7, lines 186-187: This comparison could be included in a supplementary figure.

The distortion of the EF-Tu:tRNA complex is now compared to that of the aIF2:tRNA complex in supplementary Fig. 6.

7. The authors might get a native English person to go through the manuscript. There are a number of improvements that could be made: For example, Page 8, line 193: perhaps "inaccurate" is a more appropriate term than "poorly accurate"? Page 8, line 197: perhaps "weaker" is a more appropriate term than "less strong"? Page 8, line 198: perhaps "looser" is a more appropriate term than "less tight"? etc etc...

The suggested corrections have been made and the revised manuscript has been edited by a native English speaker.

8. It would be beneficial to have Figure 1 and 2 combined with four panels and using the same angles so that the views in figure 2 can be related to the overviews in Figure 1.

As suggested by the reviewer, figures 1 and 2 have been combined with four panels using the same angles (Fig. 1 of the revised version).

9. Page 18, aIF1A mRNA? Is this the name of the mRNA, or is this mRNA encoding aIF1A? Perhaps the authors should consider renaming the mRNA if it's the former to avoid confusion.

The sequence of this synthetic mRNA corresponds to the natural start region of the mRNA encoding elongation factor aEF1A from *P. abyssi*. This mRNA was chosen because it contains a strong SD sequence. Because of the possible confusion between aEF1A and aIF1A, the mRNA is now more explicitly described (line 98 and lines 364-367 in the Methods section).

10. Page 19, 1.75h - does this mean 1h 45min?

This point has been corrected.

11. Page 19, the authors should provide a total dose for frames used in the final reconstructions.

The total dose for frames used in the final reconstruction is now indicated in Methods (line 407) and in Supplementary Table 1 (see also point 13).

12. Page 20: From 192000 particles only 12600 and 5500 remained in the final reconstructions. This is seen in Figure 1d, however, the figure is simply too small to see what all these different states are and how they differ. It would be appropriate to describe the different states in some details in the methods.

The figure has been extended to make the different states visible. Moreover, the different states in the classification are now described in the Methods section (lines 413-417).

13. The statistics for processing and refinement are hidden in the methods. It would be easier for the reader if the authors simply made a supplementary table to summarize all the numbers, like most other structural papers.

A supplementary table has been added in order to summarize the statistics.

Reviewer #2 (Remarks to the Author):

Selection of the AUG start codon during translation initiation in eukaryotic cells requires the base-by-base scanning of the mRNA leader until complementary base-pairing interactions are established between the anticodon loop of the initiator Met-tRNA_i and the start codon in the mRNA. Genetic, biochemical and structural insights have led to a model whereby the Met-tRNA_i in complex with eIF2 binds to the P site of the 40S ribosomal subunit in a P(out) conformation where it can inspect the mRNA sequence. Upon codon-anticodon pairing, the tRNA moves to a P(in) conformation and the 40S subunit adopts a closed conformation around the mRNA that blocks further scanning. Structural support for this model has come most recently from high-resolution (4-6 Å) cryo-EM structures of preinitiation complexes from yeast.

In this paper the authors provide high-resolution (5.3 and 7.5 Å) structures of preinitiation complexes from archaea. While archaeal translation shares some resemblance with eukaryotic translation with the common requirement for translation initiation factors eIF1, eIF1A and eIF2, eukaryotic translation requires additional factors including the GTPase activating protein eIF5. Moreover, selection of the start codon during archaeal translation is facilitated by a bacterial-like Shine-Dalgarno interaction between mRNA sequences preceding the start codon and sequences near the 3' end of the rRNA in the small ribosomal subunit.

In the lower resolution structure in this paper, the Met-tRNA_i is in a P(in) conformation, base-paired to the start codon. This structure resembles the structure of the yeast P(in) complex reported previously. However, in contrast to the yeast structure, the eIF2_{gamma} subunit docks on helix h44, consistent with published chemical probing studies. Unexpectedly, the N-terminal domains of eIF2_{alpha} and the C-terminal domain of eIF2_{beta} appear to disengage from the core of the structure. This apparent disengagement of eIF2_{alpha} is particularly striking as in the yeast structure this domain contacts the context nucleotides flanking the start codon. It is unclear whether these structural differences reflect different states of the eukaryotic and archaeal complexes or mechanistic differences in the mode of translation initiation between eukaryotes and archaea. Notably, the structure also visualizes the base-pairing between the rRNA and Shine-Dalgarno mRNA sequences.

The higher resolution structure is quite surprising. In this structure the anticodon loop of the tRNA_i is pivoted out of the P site and is remote from the mRNA. While this structure would be incompatible with a scanning mechanism of start codon selection, it could reflect the alternative Shine-Dalgarno mechanism used in archaea. While the authors refer to this structure as P(out), this could be confusing to readers in the translation field as this P(out) does not resemble the P(out) conformation observed in the yeast preinitiation complex structures.

We agree with the reviewer that P(out) was confusing. P_{REMOTE} is now used in the revised version.

While the paper is written clearly and the structures apparently justify the conclusions, it would be reassuring to have independent evidence to support the unexpected orientation of the Met-tRNA_i in the so-called P(out) structure as well as evidence to support the displacement of eIF2_{alpha} and eIF2_{beta} in the P(in) structure.

In order to give further credit to the unexpected orientation of the tRNA in the IC0-P_{REMOTE} structure, we have added a biochemical experiment aimed at probing the anticodon loop accessibility to the S1 nuclease. The S1 nuclease is indeed known to cleave initiator tRNAs in their anticodon loops. The P_{REMOTE} state features a tRNA having an anticodon loop expected to be more accessible to nuclease cleavage than that of a tRNA engaged in the P site. Anticodon loop accessibility was assessed in two complexes. The first one corresponds to the complex studied by cryo-EM except that the mRNA contains a CAU codon in place of the AUG start codon, in order to favor the P_{REMOTE} conformation. The second complex contains the AUG bearing mRNA, but in order to favor accommodation of the

tRNA in the P site, aIF1 was omitted. The rate of cleavage in the anticodon loop by the S1 nuclease was higher in the former complex than in the latter ($2.1 \pm 0.07 \text{ \% min}^{-1}$ vs. $1.2 \pm 0.08 \text{ \% min}^{-1}$). This result is consistent with an increased accessibility of the anticodon loop during the start codon search process, and gives therefore further credit to the existence of the P_{REMOTE} state. The experiment is reported in Supplementary Fig. 8, in the Discussion section (lines 237-249) and in the Methods section (lines 375-390).

The second point raised by the reviewer concerns the displacements of eIF2 α -D12 and eIF2 β in the P_{IN} structure. Thanks to previous structural data these two parts of aIF2 (aIF2 α -D12 and the core domain of aIF2 β) are known to be highly mobile with respect to the core rigid part of the aIF2 structure. This point was only stated in the legend of the supplementary Figure 3 of the first version of the manuscript. For better clarity, we now explain the structural organization of aIF2 in the text, line 113-118. In the present study, we observed a clear disappearance of the electron densities for aIF2 α -D12 and for the core domain of aIF2 β in IC1-P_{IN} as compared to what is observed in IC0-P_{REMOTE}. These movements are clearly related to the repositioning of the tRNA. In particular, upon IC0-P_{REMOTE} to IC1-P_{IN} transition the position of aIF1 with respect to the tRNA molecule corresponds to that observed for aIF2 β in IC0-P_{REMOTE}. As a consequence, aIF2 β cannot any longer interact with the tRNA in IC1-P_{IN} and becomes mobile (its electron density is very weak and its position was only tentatively assigned, see text pages...). Moreover, IC0-P_{REMOTE} to IC1-P_{IN} transition is also accompanied by the loss of interactions between aIF2 α and the platform, consistently with the disappearance of the electron density. Overall, the structural data support the idea that the two mobile wings of aIF2 lose most of their contacts with the rest of the initiation complex in IC1-P_{IN}. Even if their positions cannot be safely determined in the IC1-P_{IN} complex, we believe that the structural data give evidence for the mobility of these two domains during the translation initiation process.

1) line 55: "loses" instead of "looses"

This correction has been made.

2) line 100: why is excess mRNA, aIF1A and TC added to purified complexes before spotting on grids? Could the unexpected "P(out)" conformation reflect a non-functional complex?

An excess of mRNA, aIF1A and TC was added to purified complexes before spotting on grids to ensure maximal occupancy of the small ribosomal subunits by the factors. In particular, we expected that having a higher concentration of free factors when the sample is diluted before spotting would help to reconstitute full initiation complexes by factor rebinding.

Since the IC0-P_{REMOTE} structure contains the expected stoichiometry for factors, we think it unlikely that it may have resulted from the excess of factors added. Also, the fact that aIF2 γ , aIF1 and aIF1A are found at the same positions in the two complexes argue in favor of IC0 and IC1 being functional complexes. Finally, the biochemical experiment added in the revised version gives further support to the conformation of the tRNA observed in IC0-P_{REMOTE}.

3) line 134: it might be useful to cite figure 3b here

Figure 4b of the revised version (Figure 3b in the first version) is now cited line 138.

4) Figure 4 legend, line 291: "b-" should be "(c)"

We apologize for this error. It has been corrected in the revised version (legend of Fig. 5).

5) Supplementary Figure 1a legend: the last word "(right)" is confusing.

The legend of Supplementary Fig. 1 has been corrected accordingly.

Reviewers' Comments:

Reviewer #1 (Remarks to the Author):

The authors have adequately addressed my previous concerns and comments.

Reviewer #2 (Remarks to the Author):

In this revised manuscript the authors have addressed all of the points I raised in my review of their original submission.

Using the new nomenclature P-remote to describe the unprecedented orientation of Met-tRNA in their structure brings necessary clarification from the P-out conformation described in the structure of the eukaryotic complex.

To provide support for the P-remote conformation, the authors include a new supplemental figure showing that the anticodon loop of Met-tRNA is more susceptible to nuclease cleavage when the mRNA lacks a start codon. The data are consistent with the Met-tRNA undergoing reorientation from an exposed to an engaged position upon start codon recognition. However, it is not clear whether the assay would distinguish a P-remote versus P-out conformation. The question is whether the P-remote orientation is real or an artifact. The nuclease sensitivity experiment, while consistent with the P-remote orientation, might also support a modest reorientation to the eukaryotic P-out conformation as well. The authors were cautious in their descriptions of this data in the Discussion and I feel that it is an acceptable effort to address the criticism.

A second surprising result in the structure was the substantial reorientation of domains in eIF2alpha and eIF2beta between the P-remote and P-in conformations. Rather than provide additional data supporting the reorientation, the authors cite other studies on the isolated subunits as well as on the isolated eIF2 complex that show structural flexibility of these proteins. These citations support the notion that these domains of eIF2alpha and eIF2beta are flexible, but there is no independent data supporting the authors' structural studies showing reorientation of these subunits in the P-remote versus P-in orientations.

Minor Comments:

1. Line 181: there is not supplementary figure 4c
2. Line 185: do you mean to cite Supplementary Fig. 3b? This figure does not seem to support

the sentence.

3. Line 219: perhaps specifically cite supplementary figure 7b, not the whole figure

4. Supplementary Figure 7a and 7b: change “IC0-P-out” to “IC0-P-remote”

5. Line 249: change “credit” to “credence”

6. Line 273: “labeling experiments” is not a proper description. Perhaps “directed hydroxyl radical probing experiments”

7. Line 276: remove “both”

Point-by-point responses to the reviews

Cryo-EM study of start codon selection during archaeal translation initiation

Pierre-Damien Coureux, Christine Lazennec-Schurdevin, Auriane Monestier, Eric Larquet, Lionel Cladière, Bruno P Klaholz, Emmanuelle Schmitt and Yves Mechulam

Correspondence : emmanuelle.schmitt@polytechnique.edu

All issues raised by reviewer 2 have been addressed as suggested (details below).

REVIEWERS' COMMENTS:

Reviewer #1 (Remarks to the Author):

The authors have adequately addressed my previous concerns and comments.

We thank the reviewer for these positive comments.

Reviewer #2 (Remarks to the Author):

In this revised manuscript the authors have addressed all of the points I raised in my review of their original submission.

Using the new nomenclature P-remote to describe the unprecedented orientation of Met-tRNA in their structure brings necessary clarification from the P-out conformation described in the structure of the eukaryotic complex.

To provide support for the P-remote conformation, the authors include a new supplemental figure showing that the anticodon loop of Met-tRNA is more susceptible to nuclease cleavage when the mRNA lacks a start codon. The data are consistent with the Met-tRNA undergoing reorientation from an exposed to an engaged position upon start codon recognition. However, it is not clear whether the assay would distinguish a P-remote versus P-out conformation. The question is whether the P-remote orientation is real or an artifact. The nuclease sensitivity experiment, while consistent with the P-remote orientation, might also support a modest reorientation to the eukaryotic P-out conformation as well. The authors were cautious in their descriptions of this data in the Discussion and I feel that it is an acceptable effort to address the criticism.

A second surprising result in the structure was the substantial reorientation of domains in eIF2alpha and eIF2beta between the P-remote and P-in conformations. Rather than provide additional data supporting the reorientation, the authors cite other studies on the isolated subunits as well as on the isolated eIF2 complex that show structural flexibility of these proteins. These citations support the notion that these domains of eIF2alpha and eIF2beta are flexible, but there is no independent data supporting the authors' structural studies showing reorientation of these subunits in the P-remote versus P-in orientations.

We thank the reviewer for these positive comments.

Minor Comments:

1. Line 181: there is not supplementary figure 4c.
The reviewer is right; this is indeed an error. “Supplementary Figure 4b,c” has been replaced by “Supplementary Figure 4b”.
2. Line 185: do you mean to cite Supplementary Fig. 3b? This figure does not seem to support the sentence.
The reviewer is right; this is indeed an error. “Supplementary Figure 3b” has been replaced by “Supplementary Figure 4b”.
3. Line 219: perhaps specifically cite supplementary figure 7b, not the whole figure
This has been done.
4. Supplementary Figure 7a and 7b: change “IC0-P-out” to “IC0-P-remote”
This has been done.
5. Line 249: change “credit” to “credence”
This has been done.
6. Line 273: “labeling experiments” is not a proper description. Perhaps “directed hydroxyl radical probing experiments”
This has been done.
7. Line 276: remove “both”
This has been done.